# Speech-CLAP: Towards Style-Aware Speech Representation

## Abstract

Contrastive Language–Audio Pretraining (CLAP) has shown strong performance in modeling general audio–text, but remains limited in capturing complex and diverse speech styles. We propose Speech-CLAP, a contrastive model that learns joint representations of speech audio and style descriptions, capturing both intrinsic speaker characteristics (e.g., age, gender, timbre) and dynamic expressive features (e.g., emotion, speaking rate, intonation). The model is trained on a 10,000-hour speech–style corpus with detailed textual descriptions of speech styles, and we further introduce the **S**peech-**S**tyle **S**imilarity Benchmark ($S^3$Bench), the first cross-lingual benchmark for speech-style similarity, which includes both Chinese and English speech-style pairs with human preference annotations. Experimental results show that Speech-CLAP aligns closely with human judgments. This work not only provides a solid foundation for style-aware speech representation but also establishes an important evaluation standard for future research on speech-style modeling. We will release both the Speech-CLAP model and the $S^3$Bench to the community to facilitate future research on speech-style modeling.

## 1 Introduction

Contrastive learning has recently emerged as a dominant paradigm for multimodal representation learning, achieving remarkable success in computer vision and audio processing. Inspired by the breakthrough of Contrastive Language–Image Pretraining (CLIP) (Radford et al., 2021) in vision–language modeling, a series of Contrastive Language–Audio Pretraining (CLAP) (Elizalde et al., 2023) approaches have been proposed, aligning audio signals with natural language descriptions to enable general-purpose audio retrieval and classification. While these models excel in representing general audio events (e.g., dog barking, door slamming, piano playing), they remain limited when applied to human speech, where not only the content (what is said) but also the manner of speaking (how it is said) carries crucial information.

Speech style is inherently nuanced, including both intrinsic speaker characteristics (e.g., age, gender, timbre) and dynamic expressive features (e.g., emotion, speaking rate, intonation). Recent advances in controllable TTS (Guo et al., 2022; Shimizu et al., 2024; Yang et al., 2024) and style captioning (Vyas et al., 2023; Ji et al., 2024; Jin et al., 2024) highlight the promise of using natural-language prompts to describe speech style. However, many approaches are still limited by their reliance on pre-defined categories and labels—they compose style descriptions by combining discrete tags from a fixed set (e.g., emotion, speaking rate, pitch/timbre), as demonstrated in Table 1. Some recent studies attempt to move beyond such categorical tags by directly learning embeddings of specific expressive dimensions. For instance, *emotion2vec* (Ma et al., 2023) introduces a universal emotion representation via self-supervised learning, while RA-CLAP (Sun et al., 2025) augments CLAP for emotional speaking style retrieval. Other efforts such as CLAPSpeech (Ye et al., 2023) focus on mapping text to prosody embeddings. Yet these models remain confined to individual aspects such as emotion or prosody, and thus fall short of providing a general-purpose, multidimensional, and natural-language–aligned representation of speech style.

To bridge this gap, we propose Speech-CLAP, a contrastive learning model trained on a 10,000-hour corpus of speech–style pairs with fine-grained stylistic descriptions. Unlike prior CLAP variants (Wu et al., 2023) that primarily target sound events, Speech-CLAP is designed to capture the nuanced stylistic dimensions of human speech in a cross-lingual setting. Our Speech-CLAP

Table 1: Examples of style descriptions from representative datasets compared with our SPEECH-CLAP corpus.

| Dataset | Examples |
| --- | --- |
| PromptTTS | "The rapid, loud and high-keyed voice belongs to the lady." 
 "One man said loudly." |
| TextrolSpeech | "The male speaker's energetic discourse is accompanied by a normal pitch and speed." 
 "Speaking with normal energy, she conversed swiftly." |
| SpeechCraft | "With a natural emotion, a normal-pitched and normal-volume female youth speaks at a normal pace and shares a thought." 
 "青年男性以中等的音高、较高音量，以悲伤的语气慢慢地说。" |
| **Speech-CLAP (Ours)** | **"A slow, gravelly whisper conveying a sophisticated and sinister threat."** 
 "一种表面甜美却暗藏杀机的声音,逐渐转向阴森威胁的语气,是典型反派角色的特征。" |

corpus encourages free-form, fine-grained descriptions beyond discrete labels, advancing the field from label assembly toward open-text, high-resolution style representations as shown in Table 1. As a foundation representation model, SPEECH-CLAP can serve as the basis for multiple downstream tasks beyond similarity evaluation. Although this work primarily focuses on style similarity, prior studies have demonstrated the potential of style-aware representations in tasks such as controllable text-to-speech (Guo et al., 2022) and as front-end encoders for large audio–language models (LALMs) (Ghosh et al., 2025).

At the same time, rigorous evaluation of style representations remains challenging due to the lack of nuanced evaluation frameworks: existing datasets predominantly use tag-based descriptions (Guo et al., 2022; Ji et al., 2024; Jin et al., 2024) that fail to capture the richness of human-perceived speaking style. Moreover, there is no established way to measure whether learned representations truly capture the stylistic nuances that humans perceive in speech. To fill this gap, we introduce the **S**peech-**S**tyle **S**imilarity **B**enchmark ($S^3$**Bench**), the first cross-lingual benchmark that evaluates how well speech–style style representations align with human perception via human-annotated 0–5 ratings. $S^3$Bench comprises 1,000 Chinese and English speech–style pairs, each annotated by five raters (5,000 ratings total). Our experiments show that SPEECH-CLAP achieves strong correlations with human judgments (Pearson = 0.69, Spearman = 0.69, p < 1e-140). This demonstrates that our model successfully learns fine-grained speech style representations aligned with human perception.

In summary, our contributions are threefold:

- We introduce $S^3$**Bench**, the first cross-lingual human preference benchmark for evaluating fine-grained speech style similarity.
- We present SPEECH-CLAP, the first contrastive learning model explicitly designed to represent fine-grained speech styles defined by natural language across Chinese and English.
- Extensive experiments demonstrate that SPEECH-CLAP achieves high consistency with human judgments on cross-lingual style similarity assessment.

## 2 SPEECH-STYLE SIMILARITY BENCHMARK

### 2.1 MOTIVATION

Evaluating fine-grained speech style representations poses unique challenges. First, the distribution of speaking styles is inherently imbalanced, making it difficult to ensure balanced coverage across different styles. Second, natural and fine-grained style descriptions often lead to low inter-annotator agreement, as human raters may emphasize different stylistic dimensions when assessing similarity. These issues complicate the evaluation of whether a model truly captures subtle stylistic cues that humans perceive. To address these challenges, we design a stratified sampling strategy to maximize stylistic diversity across unknown speech style distributions, establish detailed annotation guidelines to ensure consistency, and verify the reliability of human ratings. The pipeline is shown in Fig. 1.

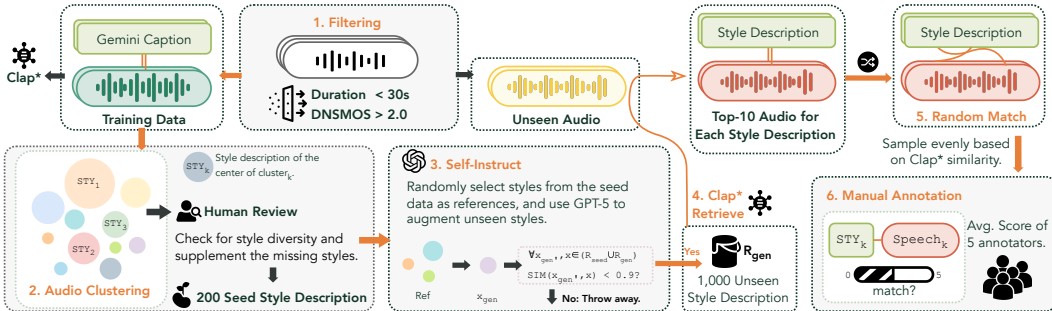

Figure 1: Construction pipeline of the Speech-Style Similarity Benchmark ($S^3$**Bench**). Here, Clap$^*$ is an intermediate model trained from training data used for retrieving audio for the benchmark.

## 2.2 BENCHMARK CONSTRUCTION

Building upon the preprocessing pipeline described in later Section 3.2, we introduce additional steps to construct a reliable benchmark. To construct $S^3$Bench, we leveraged an intermediate checkpoint of SPEECH-CLAP (denoted as Clap$^*$), obtained before the benchmark was finalized. This preliminary model was used only to provide a rough similarity signal, ensuring coverage across the similarity spectrum—from clear mismatches to close stylistic matches.(see Section 3 for details).

**Style seeds and coverage.** To ensure comprehensive coverage across stylistic dimensions and linguistic diversity, we first compute embeddings using Clap$^*$ and perform K-means clustering to group stylistically similar samples. After experiments, we set the number of clusters to 100, striking a balance between granularity and coverage without creating overly fragmented groups. The cluster centers were then manually inspected, and underrepresented categories were supplemented. Through this combination of automatic clustering and manual refinement, we ultimately obtain 200 style seeds for each of the Chinese and English subsets.

**Self-instruct expansion.** We adopt a self-instruct expansion strategy(Wang et al., 2022b), where the initial style seeds are used as prompts to iteratively synthesize diverse style descriptions. In each iteration, style descriptions are sampled at a 7:3 ratio from the seed pool and the generated pool, employing GPT-5 to produce new candidates and thus ensure scalability and diversity. The prompt is shown in Appendix B.1. To preserve diversity, for each new candidate, we use Qwen-Embedding-8B to compute its cosine similarity against every description already in the generation pool and the seed data, and admit the candidate only if all pairwise similarities were below 0.9. After several iterations, this process yield approximately 1,000 style descriptions.

**CLAP-guided stratified sampling.** We use Clap$^*$ model to retrieve candidate audio–text pairs from the corpus and the description pool, and then randomly shuffle the pairings. For each shuffled pair, we compute a Clap$^*$ similarity score $s_0$. We divide all candidate pairs into 10 bins based on the quantiles of $s_0$ and sample an equal number of pairs from each bin. This procedure yield a spectrum of pairs ranging from completely mismatched to highly matched. It is important to note that CLAP is only used in the sampling stage to balance the difficulty distribution and avoid extreme skewness. The final benchmark labels are based solely on human ratings.

## 2.3 HUMAN SIMILARITY ANNOTATION

To obtain reliable human judgments, we recruited 20 annotators, all undergraduate or graduate students from Chinese universities with strong English proficiency. Annotators were paid 1 RMB per pair, corresponding to an hourly rate of about 7–8 USD, which is above the local minimum wage. Before annotation, each annotator completed a qualification test of 10 items and is required to maintain an outlier rate below 30% to proceed. During the main task, each annotator rate 250 pairs, with every pair independently rated by five annotators. The rating scale ranges from 0 (completely mismatched) to 5 (highly matched), reflecting the perceived degree of stylistic alignment between the speech sample and the text description. This process results in 5,000 ratings in total, and the average

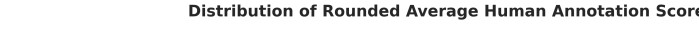

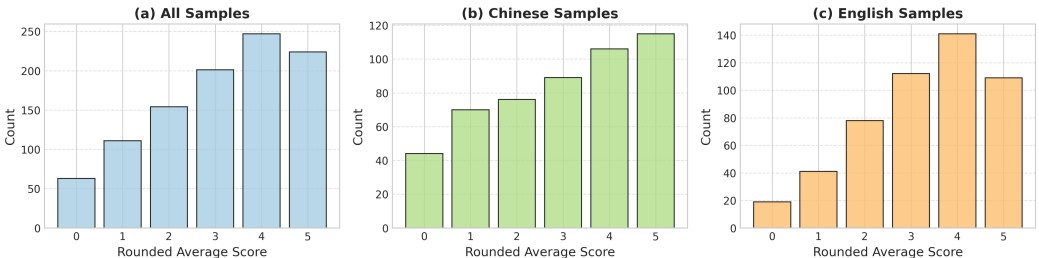

Figure 2: Average rating distribution of $S^3$Bench. Each sample is annotated with a mean score over five judgments, spanning the full 0–5 range, which reflects the benchmark's coverage of different levels of stylistic match.

rating of each pair was taken as its reference score. We ensure that annotators are only exposed to the audio and text pairs without any metadata to prevent bias. The full annotation guidelines are detailed in Appendix A.

## 2.4 DATASET ANALYSIS AND STATISTICS

We conduct a detailed analysis of the $S^3$Bench to verify its reliability and highlight its challenges. The analysis covers score distribution, inter-annotator agreement, and language-specific subsets.

**Score distribution.**   The rating distribution of $S^3$Bench spans the full range from 0 (completely mismatched) to 5 (highly matched), which is consistent with our design objective of covering pairs of varying difficulty. As shown in Figure 2, the dataset naturally provides a balanced spectrum from clear matches to clear mismatches, demonstrating that our CLAP-guided stratified sampling effectively produces a diverse benchmark across different levels of stylistic similarity.

**Inter-annotator agreement.**   We report Krippendorff's(Krippendorff, 2018) $\alpha$ (both interval- and ordinal-distance variants) for the full dataset as well as the Chinese and English subsets. Krippendorff's $\alpha$ quantifies inter-annotator reliability by comparing the observed disagreement $D_o$ with the disagreement expected by chance $D_e$:

$$\alpha = 1 - \frac{D_o}{D_e}, \tag{1}$$

where

$$D_o = \frac{\sum_c \sum_k o_{ck}\, \delta^2(c,k)}{\sum_c \sum_k o_{ck}}, \qquad D_e = \frac{\sum_c \sum_k e_c e_k\, \delta^2(c,k)}{\sum_c \sum_k e_c e_k}.$$

Here $o_{ck}$ denotes the observed co-occurrence of categories $c$ and $k$ within units, $e_c$ is the marginal frequency of category $c$, and $\delta(c,k)$ is a distance function between labels (we use squared ordinal distance for 0–5 ratings).

As shown in Table 2, using all five ratings per item (*raw*) yields an overall $\alpha = 0.50$ (Chinese: $0.59$; English: $0.41$), reflecting moderate agreement and certain language-specific variation. To examine sensitivity to extreme judgments, we further perform *symmetric trimming*: discarding the highest and lowest rating for each item before aggregation. The trimmed results substantially increase overall $\alpha$ to $0.75$ (Chinese: $0.80$; English: $0.68$), indicating strong consensus among the majority of annotators once extremes are down-weighted. More importantly, Pearson and Spearman correlations between model scores and human labels remain nearly identical when using trimmed versus raw means, demonstrating that extreme ratings have limited effect on the benchmark conclusions and that the benchmark is robust.

Robust statistics show that trimmed (or Winsorized) means improve estimation reliability under outliers or heavy-tailed noise(Lugosi & Mendelson, 2020), we therefore report both (i) the raw $\alpha$ using all ratings (our primary result) and (ii) the trimmed $\alpha$ as a robustness check, jointly supporting that the benchmark labels remain reliable and stable across aggregation choices.

Table 2: Inter-annotator agreement on $S^3$Bench. We report Krippendorff's $\alpha$ (interval/ordinal) (categorical 0–5). Both raw (all five ratings) and trimmed (discarding highest and lowest per item) results are shown.

| Dataset | $\alpha$ (raw) | $\alpha$ (trimmed) |
|---|---|---|
| $S^3$Bench (All) | 0.50 | 0.75 |
| $S^3$Bench (ZH) | 0.59 | 0.80 |
| $S^3$Bench (EN) | 0.41 | 0.68 |

In summary, $S^3$Bench provides balanced coverage across the similarity spectrum. Although the task itself is more complex, the majority of annotators still achieved a reasonably reliable level of agreement ($\alpha > 0.667$) (Krippendorff, 2018). These properties make it a trustworthy and valuable benchmark for style-aware speech representation.

## 3 SPEECH-CLAP

### 3.1 OVERVIEW

We propose SPEECH-CLAP, a strong model that extends the CLAP framework to fine-grained speech style representation. Unlike previous CLAP variants that mainly focus on general audio events, SPEECH-CLAP is designed to align natural language descriptions with human speech, emphasizing paralinguistic and stylistic attributes.

### 3.2 TRAINING CORPUS AND PREPROCESSING

Speech-CLAP is trained on a large-scale corpus of approximately 10,000 hours of speech–style pairs. To ensure quality and stylistic diversity, we designed a systematic data preprocessing pipeline:

**Data source selection.** We draw data from multiple sources, including podcasts, dubbing corpora, movie dialogues, and public speeches. These sources provide stylistically diverse material: podcasts often reflect casual conversation, while movie scripts and dubbing include highly emotional and expressive speech. Such diversity broadens the stylistic coverage of the corpus.

**Segmentation and Quality Filtering** We first applied voice activity detection (VAD) to obtain short, speech-only segments from long recordings, and we limited each segment to $\leqslant 30$ s to avoid mixing multiple styles within a single sample and to preserve stylistic consistency. Then, we evaluated audio quality with DNSMOS, adjusting the threshold to remove low-quality samples while retaining as many human- voice recordings as possible. This step reduces noise and low-fidelity artifacts while enriching the stylistic diversity of the dataset.

**Speaker Distribution** To prevent bias toward a small number of voices, we maintained a balanced proportion of single-speaker and multi-speaker samples with speaker tags generated by whisperD [1] covering a wide range of genders, ages, and accents, and including a subset of conversational speech. This design allows the model to capture both the unique characteristics of individual speakers and the stylistic patterns emerging in multi-speaker interactions.

**Caption Generation and Rewriting** On the text side, following Huang et al. (2025), we used Gemini-2.5-pro to generate multi-dimensional stylistic descriptions, covering attributes such as timbre, rhythm, emotion, and accent. To improve robustness, we further applied Qwen-8B to rewrite the captions, generating multiple variants for each speech sample. This augmentation mitigates the risk of overfitting to fixed templates and introduces stylistic paraphrase diversity into training. The prompt used in rewriting is shown in Appendix B.2.

Together, these steps constitute a preprocessing and captioning pipeline designed to ensure both high quality and stylistic diversity of the training data.

---

[1] https://huggingface.co/jordand/whisper-d-v1a

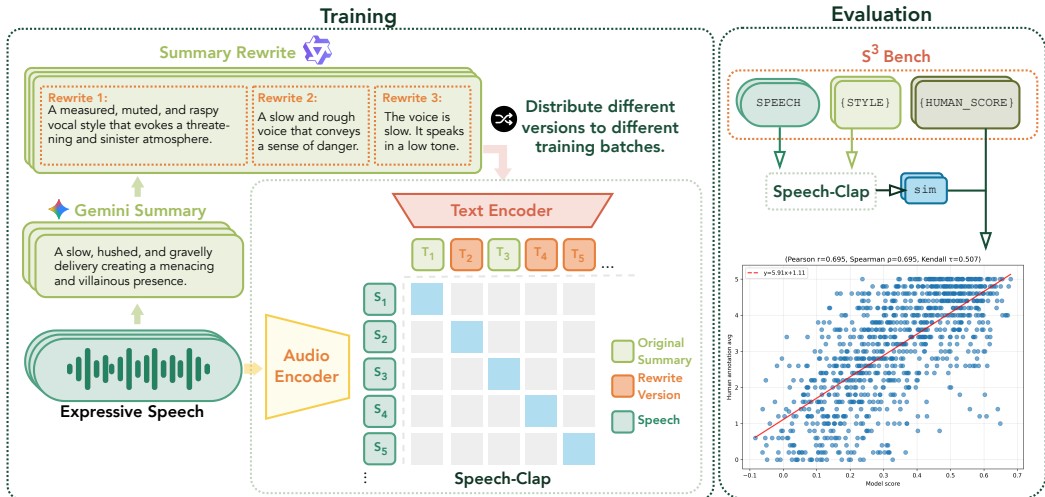

Figure 3: Illustration of the SPEECH-CLAP data pipeline. Raw speech–style pairs are captioned with multi-dimensional stylistic attributes, augmented by Qwen, and then used to train the SPEECH-CLAP dual-encoder model.

### 3.3 MODEL ARCHITECTURE

Speech-CLAP follows the dual-encoder design of CLAP. For audio encoding, we adopt HTS-AT Large(Chen et al., 2022), a spectrogram Transformer that has demonstrated strong performance on general audio understanding benchmarks and has been validated in prior CLAP frameworks. HTS-AT employs a hierarchical Transformer architecture that operates on spectrogram representations. By modeling time–frequency structures in a multi-scale manner, it provides strong performance on audio tasks. Although not specifically designed for speech prosody, its ability to capture both local and global spectro-temporal patterns makes it a suitable backbone for speech-related applications.

Considering our goal of building a multilingual and style-sensitive model, we adopt mT5Xue et al. (2020) as a strong and widely used baseline for the text encoder. In our experiments, we compared mT5 with Qwen3-Embedding-0.6B and ultimately selected the latter, as it offers balanced representation capability in both Chinese and English while maintaining a lightweight design that avoids overshadowing the audio modality.

Both audio and text embeddings are projected into a shared space, where cosine similarity is computed for contrastive alignment. The architecture remains simple and reproducible, consistent with our design philosophy.

### 3.4 TRAINING OBJECTIVE

Following LAION-CLAP, we adopt a symmetric InfoNCE contrastive loss to align audio and text embeddings. Given a batch of $N$ paired samples $\{(a_i, t_i)\}_{i=1}^N$, we compute the similarity matrix as

$$s_{ij} = \frac{\mathbf{a}_i \cdot \mathbf{t}_j}{\tau}, \tag{2}$$

where $\tau$ is a learnable temperature parameter.

The contrastive loss is defined in both audio-to-text and text-to-audio directions. For the audio-to-text direction, each audio embedding $\mathbf{a}_i$ is treated as a query and all text embeddings $\{\mathbf{t}_j\}$ as candidates. Similarly, for the text-to-audio direction, each text embedding $\mathbf{t}_i$ is matched against all audio embeddings $\{\mathbf{a}_j\}$. The final objective is the average of the two directions:

$$\mathcal{L} = -\frac{1}{2N} \sum_{i=1}^N \left[ \log \frac{\exp(s_{ii})}{\sum_{j=1}^N \exp(s_{ij})} + \log \frac{\exp(s_{ii})}{\sum_{j=1}^N \exp(s_{ji})} \right]. \tag{3}$$

This formulation is exactly consistent with the implementation in `ClipLoss`, where the similarity matrix is computed via dot products between audio and text embeddings, and the cross-entropy loss is applied symmetrically in both directions.

## 4  EXPERIMENTS

### 4.1  SETUP

We evaluate all models on $S^3$Bench. The evaluation protocol follows the semantic similarity paradigm of the STS-Benchmark, where model-predicted similarity scores are compared with human preference scores. We evaluate models using three widely adopted correlation-based metrics: Pearson correlation, Spearman correlation, and Kendall's $\tau$. Among them, Pearson and Spearman correlations serve as our primary indicators, since they directly reflect the consistency between model-predicted similarity and human ratings in both linear and rank-based perspectives. Kendall's $\tau$ is additionally reported as a complementary reference metric, offering a more conservative estimate of ordinal association. All models are evaluated on $S^3$Bench without additional fine-tuning.

### 4.2  BASELINES

We compare SPEECH-CLAP against several representative baselines:

**Random**  Two random strategies are considered: (i) *shuffle*, where human-annotated ratings are randomly redistributed across speech pairs, and (ii) *uniform*, where similarity scores are randomly sampled from a uniform distribution over $[0, 5]$.

**Text-only (Qwen3-Embedding-8B)**  To examine whether speech signals contribute beyond textual information, we adopt the strongest open-source text embedding model, Qwen3-Embedding-8B (Yang et al., 2025), to compute similarity between transcripts of speech and style descriptions.

**CLAP**  We adopt the state-of-the-art CLAP variant introduced by Elizalde(Elizalde et al., 2024), which employs the HTSAT-22 audio encoder and a GPT2-based text encoder trained on millions of audio–text pairs, representing the most advanced CLAP model tailored for general-purpose audio–text alignment.

**LAION-CLAP**  LAION-CLAP In our experiments, we compared publicly available checkpoints of LAION-CLAP, including larger clap general, larger clap music and speech, and clap-htsat-fused variants. These checkpoints differ in training data makeup and architecture. Since our benchmark emphasizes fine-grained speech style similarity, we selected larger clap music and speech as the primary baseline, as it provides stable and reliable performance on speech-related evaluation.

**AF-CLAP**  AF-CLAP represents an improved CLAP variant with stronger data augmentation and alignment strategies. It serves as a stronger general-purpose baseline for comparison.

### 4.3  RESULTS

We report Pearson correlation, Spearman correlation, and Kendall's $\tau$ of the models introduced in Section 4.2. Results are shown in Table 3, from which we notice that:

Random setting yield near-zero correlation with human judgments, establishing the lower bound of performance. The text-only baseline (Qwen) performs poorly across all languages (Pearson $\leqslant 0.18$), confirming that speech style information can't be adequately captured through textual descriptions alone. Other CLAP models (CLAP, LAION–CLAP, AF–CLAP) show moderate performance only on English data, with correlations ranging from 0.23 to 0.45, and cannot handle Chinese speech at all.

Our proposed SPEECH-CLAP model significantly outperforms all baseline methods across all evaluation metrics. On the combined dataset, SPEECH-CLAP achieves a Pearson correlation of 0.70,

Table 3: Performance comparison on $S^3$Bench. We report Pearson correlation (all/zh/en), Spearman correlation (all/zh/en), and Kendall $\tau$ (all/zh/en). Best results are highlighted in **bold**. Models marked with * do not support Chinese, and thus their zh results are close to random, indicating that they fail to capture meaningful cross-lingual representations.

| Model | Pearson | | | Spearman | | | Kendall $\tau$ | | |
|---|---|---|---|---|---|---|---|---|---|
| | all | zh | en | all | zh | en | all | zh | en |
| Random | 0.05 | 0.06 | 0.04 | 0.03 | 0.06 | 0.06 | 0.02 | 0.04 | 0.04 |
| Text-only (Qwen) | 0.15 | 0.18 | 0.11 | 0.15 | 0.17 | 0.12 | 0.10 | 0.12 | 0.08 |
| CLAP* | 0.16 | -0.07 | 0.45 | 0.13 | -0.09 | 0.42 | 0.09 | -0.07 | 0.29 |
| LAION-CLAP* | 0.21 | 0.04 | 0.38 | 0.19 | 0.03 | 0.36 | 0.13 | 0.02 | 0.25 |
| AF-CLAP* | 0.18 | 0.01 | 0.26 | 0.16 | 0.00 | 0.23 | 0.11 | 0.00 | 0.16 |
| Speech-CLAP (Ours) | **0.70** | **0.74** | **0.64** | **0.69** | **0.73** | **0.64** | **0.51** | **0.53** | **0.46** |

substantially higher than the best existing baseline (CLAP at 0.45 on English-only data). SPEECH-CLAP also maintains consistently strong performance across both Chinese and English subsets, with Pearson correlations of 0.74 and 0.64 respectively.

In conclusion, our SPEECH-CLAP demonstrates strong consistency with human evaluation, being the first model able to handle Chinese speech style modeling while achieving superior performance across both languages.

## 4.4 ABLATION STUDY

To better understand the contribution of different design choices, we conduct an ablation study focusing on the effect of caption rewriting. As shown in Table 4, starting from an mT5-based text encoder, replacing it with Qwen improves cross-lingual style representation. Further incorporating Qwen-8B rewrites consistently enhances correlation with human ratings, suggesting that diverse stylistic paraphrases strengthen model robustness and generalization.

**Case Study** To further illustrate the diversity and reliability of $S^3$Bench, we highlight several representative cases where style captions involve fine-grained and multi-dimensional descriptions. Despite their complexity, human annotators consistently reached high agreement, and SPEECH-CLAP also assigned high similarity scores.

It is worth noting that SPEECH-CLAP similarity scores in our setting generally fall between 0 and 0.6, consistent with distributions observed in clipscore(0-0.4) (Hessel et al., 2022). Within this range, a score above 0.5 indicates a strong stylistic match, which aligns well with the unanimous or near-unanimous human annotations in these representative cases.

These cases show that $S^3$Bench supports highly detailed and diverse style descriptions with reliable human consensus, and that SPEECH-CLAP effectively captures such fine-grained stylistic cues.

Table 4: Ablation study on the effect of text encoder choice and caption rewriting. Results are reported on $S^3$Bench.*Trimmed results (removing max/min ratings before averaging); see Table 2 for details.

| Text encoder | Pearson | | | Spearman | | | Kendall $\tau$ | | |
|---|---|---|---|---|---|---|---|---|---|
| | all | zh | en | all | zh | en | all | zh | en |
| mT5 | 0.42 | 0.44 | 0.39 | 0.41 | 0.43 | 0.38 | 0.29 | 0.30 | 0.27 |
| Qwen(Gemini) | 0.66 | 0.69 | 0.62 | 0.65 | 0.68 | 0.61 | 0.46 | 0.48 | 0.44 |
| Qwen(Rewrite) | **0.70** | **0.74** | **0.64** | **0.69** | **0.73** | **0.64** | **0.51** | **0.53** | **0.46** |
| Qwen(trimmed)* | 0.69 | 0.73 | 0.65 | 0.70 | 0.73 | 0.65 | 0.52 | 0.54 | 0.48 |

Table 5: Representative case studies with complex style descriptions. Human annotations show high agreement, and SPEECH-CLAP achieves consistent similarity scores.

| Style Caption | Human Annotations | Speech-CLAP Similarity |
|---|---|---|
| 声线轻柔如水，缓缓诉说着内心的挣扎与疑惑，语气中夹杂着未解的心结与迷茫。 | [4, 5, 5, 5, 5] | 0.52 |
| An animated children's storyteller bounces between characters with bright, playful inflection and a buoyant sing-song rhythm. | [5, 5, 4, 4, 5] | 0.53 |

## 5 RELATED WORK

### 5.1 SPEECH STYLE MODELING AND CONTROLLABLE TTS

Modeling speech style is a long-standing challenge. Traditional approaches rely on categorical emotion recognition (Feng & Narayanan, 2023) or speaker identification (Wang et al., 2022a), but these categorical labels cannot capture the full richness of human speaking style. To address this, recent work has explored free-form natural language descriptions. PromptTTS (Guo et al., 2022) introduces natural language prompts to control text-to-speech generation, while PromptTTS++ (Shimizu et al., 2024) extends this approach by combining style and speaker descriptions. StyleCap (Yamauchi et al., 2022) and SpeechCraft (Jin et al., 2024) generates multi-dimensional speech style captions, and ParaSpeechCaps (Diwan et al., 2025) provides large-scale labeled data with 59 stylistic attributes. However, these works mostly generate style descriptions based on pre-defined categories, limiting their abilities to capture the fine-grained and nuanced characteristics of natural speech.

### 5.2 CONTRASTIVE LANGUAGE–AUDIO PRETRAINING

Inspired by CLIP (Radford et al., 2021), contrastive language–audio pretraining (CLAP) has recently gained attention in audio understanding. CLAP models (Elizalde et al., 2023; Wu et al., 2023) align audio recordings with natural language descriptions in a shared embedding space. Extensions such as AF-CLAP (Ghosh et al., 2025) and M2D-CLAP (Niizumi et al., 2024) further improve audio–text alignment via enhanced data augmentation or modality fusion. However, these models primarily target general audio events (e.g., environmental sounds), and only limited attempts have been made to capture human speech style. A notable step forward is RA-CLAP (Sun et al., 2025), which introduces relation-augmented training for emotional speaking style retrieval, but it remains constrained by open-source datasets with coarse labels.

### 5.3 HUMAN PERCEPTION AND SIMILARITY BENCHMARKS

Evaluating model alignment with human perception has been studied. In natural language processing, the STS-Benchmark (Cer et al., 2017) measures semantic similarity via human ratings on sentence pairs. In vision–language research, CLIPScore (Hessel et al., 2022) provides a reference-free metric correlated with human judgment for image captioning. Recently, Human-CLAP (Shinohara et al., 2024) highlighted that original CLAP scores correlate poorly with human evaluations, and proposed human-annotated fine-tuning to bridge the gap. These studies demonstrate that human preference alignment is essential for evaluating multimodal representation models.

## 6 CONCLUSION

In this paper, we introduced SPEECH-CLAP, the first CLAP-style model explicitly designed for fine-grained speech style representation, and proposed the Speech-Style Similarity Benchmark ($S^3$Bench), the first human preference benchmark for evaluating speech–style alignment. Our experiments show that our $S^3$Bench achieves high inter-annotator agreement, validating $S^3$Bench as a reliable evaluation protocol. SPEECH-CLAP achieves strong correlation with human judgments, validating the feasibility of modeling speech style via contrastive learning.

## ETHICS STATEMENT

This work adheres to the ICLR Code of Ethics. In this study, no human subjects or animal experimentation was involved. All datasets used, including our $S^3$Bench, were sourced in compliance with relevant usage guidelines, ensuring no violation of privacy. We have taken care to avoid any biases or discriminatory outcomes in our research process. No personally identifiable information was used, and no experiments were conducted that could raise privacy or security concerns. We are committed to maintaining transparency and integrity throughout the research process.

## REPRODUCIBILITY STATEMENT

We have made substantial efforts to ensure the reproducibility of our work. The proposed benchmark consisting of 5,000 human-annotated samples will be made publicly available, and the baseline model together with its training and evaluation code will also be released upon acceptance. Detailed descriptions of data collection, preprocessing steps, and evaluation protocols are provided in the main paper. We believe these resources and explanations will enable the community to faithfully reproduce our results and further build upon them.

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

## A  ANNOTATION GUIDELINES

1. Please ignore factors unrelated to style, such as audio quality or truncation. Focus only on the degree to which the audio description matches the actual audio content.

2. If some English expressions are uncommon, you may use Google Translate for reference (plugin address: `https://chromewebstore.google.com`).

3. Ignore additional metaphorical scene descriptions that are hard to judge, such as "as if narrating a long-tested story."

4. For role-playing descriptions such as "retail clerk" or "comedian," ignore the textual role and judge only by listening.
   - Roles like "retail clerk" or "researcher" that do not imply a specific voice style can be ignored.
   - Roles like "comedian" or "singer" that explicitly imply stylistic vocal features should be considered.

5. Score the degree of match between the audio and the description from 0 to 5 according to the following standards:
   (a) Regular procedure: identify several key features from the text, then check if they match the speech. Give reasons based on proportion, and then assign a score accordingly.
      i. Reference dimensions:
         A. Gender, age, pitch, speech rate, volume, emotion, tone, accent, texture (e.g., hoarse), clarity, fluency
   (b) Proportional considerations:
      i. Each feature has equal weight in principle, but adjustments can be made based on listening judgment.
      ii. The degree to which the feature matches.
      iii. The duration for which the feature matches.

6. Assign a confidence level to your own score:
   (a) High: Confident that the annotation is reliable.
   (b) Medium: Some important dimensions cannot be judged; the score is less certain (for abstract or vague descriptions, less attention is needed).
   (c) Low: The description and audio are difficult to judge; the score is highly uncertain.

The detailed criteria and examples are shown in the table below. Each example was originally paired with a reference audio clip; since audio cannot be presented here, only the textual description and the scoring rationale are provided.

Table 6: Scoring Table for Audio–Text Matching

| Score | Caption Description | Example Description | Reason |
|---|---|---|---|
| 5 | **Description fully matches the audio across all dimensions.**
• 100% of the content is accurate | A clear and straightforward male narrator, emphasizing technical terms, orderly and restrained yet slightly excited. | All features match. |
| | | A passionate commentator with explosive pitch variation and dense speech rhythm, full of exaggerated exclamations and strong beat sense. | All features match. |
| | | An elderly woman with a low, hoarse yet gentle voice, slowing down her speech, elongating sentence endings, expressing weary but firm concern. | All features match. |
| | | | Continued on next page |

**Table 6 – continued from previous page**

| Score | Caption Description | Example Description | Reason |
|---|---|---|---|
| 4 | **Most dimensions match, with only 1–2 minor deviations.**
• Minor differences do not affect overall impression
• Speech rate variations not obvious
• About 70%–90% accurate | Street-rap style fast speech, sharp consonants, strong rhythm, stubborn tone. | Overall impression fits rap style, though text content is unrelated. |
| | | A mild and low voice tinged with hoarseness, calm yet firm, as if narrating a long-tested story. | "Low" but not "slow." |
| | | Storyteller-like rhythm, low-pitched with laughter, ending with humorous twists. | Storytelling correct, but no drumbeat rhythm. |
| 3 | **Core features match, but clear descriptive differences exist.**
• Major traits (gender, age group, overall style) are correct
• About 40%–70% accurate | A professional announcer reading news in standard Mandarin, with authority and fairness in tone. | First half matches, latter half does not. |
| | | A woman's voice showing dramatic variation, from deep sadness rising to passionate joy, as if performing an emotional storm. | The "rising from sadness to joy" part is inaccurate. |
| | | A nightclub female singer's hoarse humming, slight laughter, lazy elongated endings. | "Female singer" not accurate, other traits correct. |
| 2 | **Only a few features match.**
• Gender/rough age may match
• Most detailed descriptions do not match
• About 20%–40% accurate | Solemn and slow speech, carefully narrating a grand ceremony, with respect and seriousness in tone. | Second half matches. |
| | | A dessert shop clerk's soft and sticky recommendation, cheerful rising intonation, every line wrapped in sweetness. | Some "soft" and "rising intonation" fit, but overall impression is sarcastic/ironic. |
| 1 | **Only a minimal number of features match.**
• At most 1–2 traits align, overall style is very different
• About 10%–20% accurate | A youthful bright voice, slightly surprised and excited, ending with a light jump. | Only "slightly surprised and excited" applies. |
| | | A powerful and forceful tone, thunder-like, full of determination in every word. | Briefly "powerful," otherwise mismatch. |
| | | A husky female cabaret voice, airy and lazy downward endings, ambiguous and relaxed. | Only "female" correct, others mismatch. |
| 0 | **No match between description and audio across all dimensions.**
• 0% of the content is accurate | A soft and warm female voice, whispering gently, soothing like spring sunshine, delivering encouragement tenderly. | "Female whispering" incorrect. |
| | | A frank and direct voice, slow but firm rhythm, full of undeniable confidence. | Entirely incorrect. |

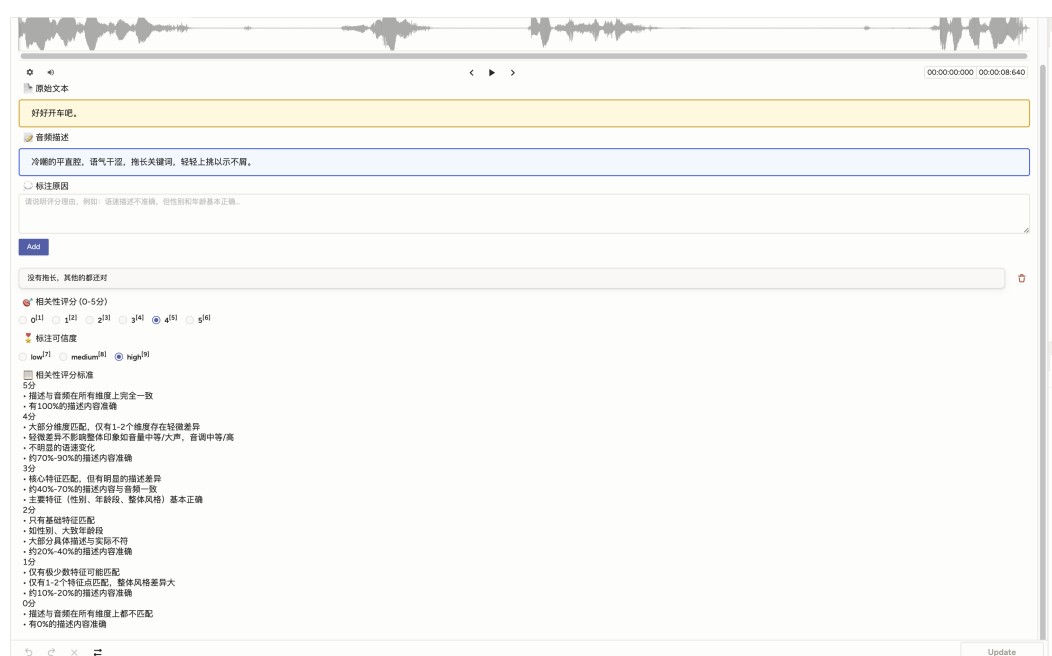

Figure 4: Annotation interface used in our study.

# B PROMPTS

## B.1 SELF-INSTRUCTING PROMPT

The following prompt is used for GPT-5 self-instructing:

```
You are a voice description data generation expert. Below are some voice
    description examples, generate more different English voice
    descriptions:

{examples}

Please continue generating voice descriptions with subsequent numbering.
    Each description should:

- Be completely new and unique voice styles
- Avoid repetition or excessive similarity with the above examples
- Descriptions can vary in length and detail
- Avoid containing too many changes and details
- Use natural Chinese expressions
- Reduce generation of bland voice styles
- Reduce descriptions starting with A or An

Start generating directly:
```

## B.2 REWRITING PROMPT

The following prompt is used for rewriting:

```
Please perform **progressive processing** on the following speech
    description text with the following requirements:

1. **Core constraint**: The original meaning must be fully preserved
    without deviation.
2. **Temporal constraint**: The chronological order of all events must
    strictly follow the original text and cannot be altered.
3. **Progressive processing**: Generate three versions of the text with
    different levels of detail, derived step by step from the original.
4. **Quality requirement**: The processed texts should be natural in
    language and logically clear.

Original text:
{summary_text}

Please provide three versions in **JSON format**:
```json
{
  "text1": "Restated version: Retain the complete meaning and level of
      detail of the original text, but rewrite it using different
      vocabulary and expressions.",
  "text2": "Simplified version: More concise than the restated version,
      focusing on the main descriptive content. Preserve the core
      information while omitting minor details, especially those that
      cannot be inferred solely from sound.",
  "text3": "Basic version: Even simpler than the simplified version,
      using short and straightforward sentences to highlight the key
      sound features."
}
```

Please ensure:
- **text1**: Same meaning as the original, same chronological order, same
    level of detail, but with different wording.
```

```
- **text2**: Extracts the main information from the original, removes
    less important details.
- **text3**: Uses simple sentences to describe the main sound
    characteristics.
```

## C USE OF LLMs

We use large language models to enhance clarity, refine style, and improve the overall quality of writing.

LLMs was not involved in the ideation, research methodology, or experimental design. All research concepts, ideas, and analyses were developed and conducted by the authors.

The authors take full responsibility for the content of the manuscript, including any text generated or polished by the LLM.

