# OpenReview forum: "Speech-CLAP: Towards Style-Aware Speech Representation"
_ICLR.cc/2026/Conference — Submitted to ICLR 2026_

### Official Review · Reviewer_B5MN · 2025-10-27

**Soundness:** 2
**Presentation:** 3
**Contribution:** 1
**Rating:** 2
**Confidence:** 5

**Summary:**

This paper introduces Speech-CLAP, a contrastive learning framework designed to generate style-aware representations for speech audio, aligning these with fine-grained and natural language style descriptions. To support robust evaluation, the authors also present the Speech-Style Similarity Benchmark ($S^3$ Bench), a new cross-lingual dataset with human-annotated judgments of speech-style alignment, spanning both Chinese and English data. Speech-CLAP is trained on 10,000 hours of speech paired with detailed style descriptions, and is shown to outperform existing CLAP variants and related baselines by a significant margin in correlation with human perceptual similarity.

**Strengths:**

1. The paper provides a new cross-lingual benchmark ($S^3$ Bench) for evaluating fine-grained speech-style similarity, with detailed annotation procedures and robust evidence of inter-annotator reliability.

2. The authors extensively analyze dataset diversity, score distribution, and inter-annotator agreement, providing transparency and reliability.

3. Speech-CLAP achieves substantially higher correlations with human ratings than prior CLAP and text-only models.

**Weaknesses:**

1. The comparison in Table 3 is unfair. You trained your own model on selected data and then tested it on a domain that closely matches your training set. However, you used off-the-shelf checkpoints for the baseline models without retraining it. This makes it unclear whether your method is truly superior, or if the performance gain simply comes from a domain mismatch.

2. The source of the training data is not disclosed; only the total duration is mentioned.

3. As a CLAP-type model, it is unacceptable that there are no retrieval results. A paper like this should be rejected!

4. The Speech-CLAP audio encoder’s superiority is not demonstrated on other tasks, such as controllable TTS or emotion recognition.

**Questions:**

See Weaknesses.

---

### Official Review · Reviewer_verL · 2025-10-27

**Soundness:** 2
**Presentation:** 3
**Contribution:** 2
**Rating:** 2
**Confidence:** 3

**Summary:**

This paper proposes a Speech-CLAP as a contrastive language-audio pretrain model which is to capture the complex nuances of human speaker characteristics (e.g. age, gender, timbre) and speech style (e.g., emotion, timbre, intonation, speaking rate). In order to evaluate the speech-style similarity, the paper further introduce the benchmark evaluation as Speech-Style Similarity Benchmark (S3 Bench).

The major contributions are the 1) construction of S3 Bench which consists 1000 audio-text pairs with both English and Chinese with human annotations on a 0-5 scale of stylistic alignment; 2) a Speech-CLAP model to learn the joint representation of speech audio and fine-grained natural language style description. Experiments demonstrate the similarity score align closely with human judgements.

**Strengths:**

The creation of the S3 benchmark for cross-lingual based on human preference is the main contribution of this paper. The design involves variations, coverage, expansion, and filtering to ensure the benchmark includes a balanced examples and carefully calculated human score with consideration of both interval and ordinal variants.

**Weaknesses:**

Major weaknesses are
1) There isn't any downstream task to apply this Speech-CLAP model to prove the effectiveness in model perspective. It only validates the similarity correlation via this S3 Benchmark, but it is not enough to prove the representation could work on any speech style generation/caption model.

2) The model architectural of speech-CLAP is lack of novelty. the dual encoder design is a standard CLIP like model, with existing components such as HTS-AT for audio and Qwen3-embedding for text, InfoNCE as contrastive loss. The effectiveness comparing with other model could come from the high-quality data.

3). The S3 bench seems to be human annotation dependent system, which means it could only evaluate the existing test set as this 1000 text-audio pair. This would be a limitation it's ability to represent the full and diversity of human speech, or other languages.

**Questions:**

1. Since it use Gemini model for audio caption, does it evaluate the caption result comparing with Gemini?

2. for the S3 Bench, it claims as cross-lingual human preference benchmark. How would it reflect the cross-lingual capability? Could it be evaluated from one language audio with another language text? how about some unseen languages?

---

### Official Review · Reviewer_6NRy · 2025-11-01

**Soundness:** 2
**Presentation:** 3
**Contribution:** 2
**Rating:** 4
**Confidence:** 4

**Summary:**

The paper presents SPEECH-CLAP, a model for learning fine-grained speech style representations from audio and text. To facilitate evaluation, the authors build S³Bench, a cross-lingual benchmark for speech style similarity with human annotations. Experiments show SPEECH-CLAP aligns well with human judgments and outperforms existing CLAP-based baselines.

**Strengths:**

The paper contributes S³Bench, a valuable and well-constructed benchmark for the under-resourced task of speech style evaluation. The construction process is rigorous, covering data filtering, clustering, self-instruct expansion from 200 seed styles to 1000 unseen styles, and manual annotation, with a detailed analysis of inter-annotator agreement.
The fine-grained modeling of speech style, which moves beyond traditional discrete labels, is of significant research value.

**Weaknesses:**

1. Writing: The paper's organization is confusing. The abs and intro prioritize SPEECH-CLAP as the core innovation without mentioning the bench until the last paragraph, yet Sec.2 abruptly shifts to the construction of the S³Bench benchmark. The Clap* in Fig 1 (which is used to build the benchmark) is unclear at this stage, hindering comprehension.
2. Clarification: The workflow for building S³Bench (Fig 1 and Sec 2.2), particularly the sampling method for the final 1,000 audio-text pairs, is not clearly described. In Step 4, the model retrieves the top 10 audio candidates for each description. However, the subsequent "randomly shuffle" and stratified sampling steps are confusing. Does this procedure imply that the initially best-matched audio samples might have been discarded? The criteria for selecting the final audio pair are ambiguous, weakening the logical chain of the benchmark's construction.
3. Limited Novelty: The fact that SPEECH-CLAP outperforms general-purpose CLAP models on a speech style task is self-evident, as it was specifically trained for this task on a large speech-style corpus. While effective, the model's architectural novelty is relatively limited, as it largely follows the established dual-encoder and contrastive learning paradigm of existing CLAP frameworks.
4. Insufficient Baseline: The performance comparison in Tab 3 lacks comparisons with more powerful models. For instance, the sota Large Audio Language Models (LALMs) or advanced closed-source multimodal models (Gemini 2.5 Pro) that possess strong speech understanding capabilities. In a task requiring nuanced understanding of both speech and text, such comparisons are crucial for comprehensively evaluating the model's performance and necessity.
5. Case Study: No reference to Tab 5 in the case study section. Besides, a robust case study should present both examples where the pair aligns well (high agreement) and where it aligns poorly (low agreement) to provide a more intuitive understanding. The lack of audio demos limits the persuasiveness of the paper.
6. Potential Circularity in Benchmark Construction: The methodology relies on an intermediate model, CLAP*, to guide the sampling for S³Bench. Although the bench is based on human scoring, it introduces a potential circular argument, as the benchmark is created with a tool similar to the model being evaluated. This risks embedding CLAP*'s biases into the benchmark, which may unfairly advantage the final SPEECH-CLAP model.

**Questions:**

1. Could the authors clarify the details of the sampling stage in S³Bench? How was the final paired audio selected? How does this process ensure both the quality and diversity of the matches in the benchmark?
2. Since the capability of SPEECH-CLAP ( and CLAP* ) in aligning speech style and text is something to be proofed, the usage of CLAP* in the construction of benchmark is somehow an circular argument. What was the precise role of CLAP*? Can you provide an comparison exchanging all CLAP*-involved procedures to a fully random sampling baseline? This would justify the necessity of CLAP* and the effectiveness of your construction methodology.
3. What is the justification for this specialized model over general-purpose Large Audio Language Models? How well does it generalize to other tasks?
4. Could you expand the case study with failure examples and provide audio demos to better illustrate the model's capabilities and limitations?

---

### Official Review · Reviewer_jvy9 · 2025-11-07

**Soundness:** 2
**Presentation:** 3
**Contribution:** 2
**Rating:** 2
**Confidence:** 4

**Summary:**

This paper proposes a new CLAP-based model for matching speech to style descriptions (that capture nuanced speech styles like emotion, timbre, etc.). Because there are no existing evaluation benchmarks and models for this task, the authors introduce (a) S3Bench, a Chinese-English benchmark created with a final human annotator step and balanced in difficulty (b) Speech-CLAP, a model trained on a Gemini-annotated synthetic dataset of 10000 hours. The results show that their models outperforms general audio CLAP models and obtains strong correlations on S3Bench.

**Strengths:**

1. The evaluation benchmark (S3-Bench) is a useful artifact. It covers two languages and was constructed with full human validation, balanced with difficult and easy examples.
2. The proposed model obtains very high correlations on this benchmark, implying it is useful.
3. The paper is well-written and easy to follow.

**Weaknesses:**

1. None of the baselines compared (Table 3) have been trained on speech-style prompt pairs; they cannot perform well on this task! Claiming that Speech-CLAP obtains SoTA results while only comparing to these baselines is misleading. The baselines are general audio CLAP models that have (a) seen much less speech data, mostly audio data and (b) are trained on speech-transcript pairs, not speech-style prompt pairs. The authors should compare with more relevant baselines that have trained on speech data. For example, ParaCLAP (https://arxiv.org/abs/2406.07203) and https://arxiv.org/abs/2508.11187 both train CLAP models with emotion style prompts.
2. The training dataset style descriptions are created synthetically simply by passing expressive speech through Gemini-2.5-Pro. There is no analysis of whether this is appropriate e.g. whether the quality of Gemini annotations are good!
3. The evaluation dataset proposed by the authors (S3-Bench) uses an intermediate checkpoint of the same Speech-CLAP model they are proposing; this is scientifically inappropriate! Using your own model in the pipeline will skew the distribution of the evaluation dataset in your model’s favor. Even if the final dataset is fully human-validated, the data distribution has been decided using CLAP-guided stratified sampling, which makes the evaluation distribution skewed towards the CLAP model distribution! This is a major experimental flaw in my opinion.
4. While the S3Bench benchmark part of the paper is novel, the training dataset is distilled from Gemini and the model architecture is identical to previous CLAP models, limiting the overall novelty of the paper.

**Questions:**

1. Why do you use the HTSAT audio encoder from the original CLAP paper rather than a stronger speech encoder? The CLAP model in AudioBox from 2023 (https://arxiv.org/pdf/2312.15821)  that is trained on speech data replaces HTSAT with WavLM as HTSAT is an audio tagger and not well-suited for speech. While not critical for this paper, I recommend that the authors try experiments with WavLM to see if the performance can be improved.
2. Can this be extrapolated to more languages beyond Chinese and English? Which languages does Gemini 2.5 Pro support for such synthetic data creation?
3. The results in table 3 show that correlation on Chinese is higher than English. Do you have any insights as to why this may be the case?

---

### Meta-Review · Area_Chair_6P3K · 2026-01-06

**Summary:**

This paper presents Speech-CLAP, a contrastive learning model for speech-style representation, along with S³Bench, a cross-lingual benchmark for evaluating speech-style similarity. While the benchmark contribution is valuable, reviewers raised significant concerns about experimental methodology, limited novelty, and potential circular evaluation that undermine the paper's claims.

**Reviewer Concerns:**

The reviewers collectively identified several critical issues that were not adequately addressed:

Unfair Baseline Comparisons: Multiple reviewers noted that comparing Speech-CLAP against general audio CLAP models that were never trained on speech-style pairs is fundamentally unfair [Reviewers jvy9, B5MN]. The reviewer may acknowledge that relevant baselines like ParaCLAP and emotion-style CLAP models were not included, making the claimed state-of-the-art results misleading.

Circular Benchmark Construction: A major methodological flaw is the use of an intermediate CLAP* model to guide S³Bench sampling [Reviewers jvy9, 6NRy]. This creates potential bias favoring Speech-CLAP in evaluation, even with human validation, as the data distribution was shaped by a similar model.

Missing Retrieval Results: For a CLAP-type model, the absence of retrieval evaluation results is a significant omission [Reviewer B5MN]. This is a standard evaluation paradigm for such models.

Limited Downstream Validation: The model is only validated on S³Bench correlation without demonstrating effectiveness on downstream tasks such as controllable TTS or emotion recognition [Reviewers verL, B5MN].

Training Data Quality Unverified: The training data relies on Gemini-2.5-Pro annotations without analysis of annotation quality or appropriateness [Reviewer jvy9].

**Reviewer Scores:**

Reviewer jvy9: 2. Justification: The reviewer may maintain concerns about limited baselines, circular benchmark construction using CLAP*, and unverified synthetic training data quality: none comprehensively addressed in rebuttal.

Reviewer verL: 2. Justification: The reviewer may remain unconvinced given lack of downstream task validation, limited novelty, and benchmark's dependency on human annotation without broader applicability demonstrated.

Reviewer 6NRy: 4. Justification: The reviewer may slightly improve given benchmark value, but concerns about confusing presentation, unclear sampling methodology, and insufficient baselines persist.

Reviewer B5MN: 2. Justification: The reviewer expressed strong conviction that missing retrieval results and undemonstrated superiority on other tasks are unacceptable for a CLAP-type paper.

---

### Decision · Program_Chairs · 2026-01-26

Reject